# Prognostic Value of miRNAs in Head and Neck Cancers: A Comprehensive Systematic and Meta-Analysis

**DOI:** 10.3390/cells8080772

**Published:** 2019-07-25

**Authors:** Chellan Kumarasamy, Madurantakam Royam Madhav, Shanthi Sabarimurugan, Sunil Krishnan, Siddhartha Baxi, Ajay Gupta, K M Gothandam, Rama Jayaraj

**Affiliations:** 1North Terrace Campus, University of Adelaide, Adelaide, South Australia 5005, Australia; 2School of Bio Science and Technology, Vellore Institute of Technology (VIT), Vellore 632014, India; 3Department of Radiation Oncology, Division of Radiation Oncology, The University of Texas MD Anderson Cancer Center, 1515 Holcombe Blvd, Houston, TX 77030, USA; 4John Flynn Private Hospital, Genesis Cancer Care, 42 Inland Drive, Tugun, Queensland 4224, Australia; 5Medical Oncology P-41, South Extension Part 2, New Delhi 110049, India; 6College of Health and Human Sciences, Charles Darwin University, Yellow 1.1.05, Ellengowan Drive, Casuarina, Darwin, Northern Territory 0909, Australia

**Keywords:** microRNAs, head and neck cancer, prognosis, systematic review, meta-analysis

## Abstract

Head and Neck Cancer (HNC) is the sixth most common type of cancer across the globe, with more than 300,000 deaths each year, globally. However, there are currently no standardised molecular markers that assist in determining HNC prognosis. The literature for this systematic review and meta-analysis were sourced from multiple bibliographic databases. This review followed PRISMA guidelines. The Hazard Ratio (HR) was selected as the effect size metric to independently assess overall survival (OS), disease-free survival (DFS), and prognosis. Subgroup analysis was performed for individual highly represented miRNA. A total of 6843 patients across 50 studies were included in the systematic review and 34 studies were included in the meta-analysis. Studies across 12 countries were assessed, with China representing 36.7% of all included studies. The analysis of the survival endpoints of OS and DFS were conducted separately, with the overall pooled effect size (HR) for each being 1.825 (95% CI 1.527–2.181; *p* < 0.05) and 2.596 (95% CI 1.917–3.515; *p* < 0.05), respectively. Subgroup analysis was conducted for impact of miR-21, 200b, 155, 18a, 34c-5p, 125b, 20a and 375 on OS, and miR-21 and 34a on DFS. The pooled results were found to be statistically significant for both OS and DFS. The meta-analysis indicated that miRNA alterations can account for an 82.5% decrease in OS probability and a 159.6% decrease in DFS probability. These results indicate that miRNAs have potential clinical value as prognostic biomarkers in HNC, with miR-21, 125b, 34c-5p and 18a, in particular, showing great potential as prognostic molecular markers. Further large scale cohort studies focusing on these miRNAs are recommended to verify the clinical utility of these markers individually and/or in combination.

## 1. Introduction

Head and Neck Cancer (HNC) is the sixth leading cancer by incidence worldwide, with an annual incidence rate of more than 550,000 cases and around 300,000 deaths each year [1]. It has shown to be strongly associated with environmental and lifestyle risk factors including tobacco use, alcohol consumption, human papilloma virus or Ebstein-Barr virus infection, poor oral and dental hygiene, malnutrition, gastroesophageal or laryngopharyngeal reflux disease, and occupational exposure to chemicals and fumes [2]. Treatment for early HNC primarily involves multi-modality therapy with a combination of surgery and radiation [3]. Patients with advanced disease stages have been shown to frequently develop recurrences or distant metastases, resulting in five-year overall survival (OS) rates of less than 60% with poor long-term prognosis [4]. Patients with loco-regional relapse or metastatic disease usually cannot be cured, with only limited benefits arising from palliative chemotherapy [5]. In this regard, there is an urgent need for the identification of biomarkers for accurate prognosis, in order to inform and direct patient treatment. MicroRNAs (miRNAs) have been proposed as quantitative prognostic markers in HNC [6]. MicroRNAs are small non-coding RNAs (~22 nucleotides) transcribed from DNA into RNA hairpins. MicroRNAs post-transcriptionally regulate gene expression by binding to the 3′-UTR of target mRNAs, resulting in target mRNA degradation or inhibition of their translation [7,8]. Other recent studies have also suggested that the secondary structure of the 5′ untranslated region (5′ UTR) of messenger RNA (mRNA) is important for microRNA (miRNA)-mediated gene regulation in humans [9]. MiRNAs are first transcribed as primary transcripts (pri-miRNA) with a cap and poly-A tail by RNA polymerase II or RNA polymerase III, with a typical pri-miRNA being composed of a double stranded stem of ’33 base pairs, a terminal ‘hairpin’ loop and two flanking unstructured single-stranded segments. MiRNAs are first transcribed as primary transcripts (pri-miRNA) with a cap and poly-A tail by RNA polymerase II or RNA polymerase III, with a typical pri-miRNA being composed of a double stranded stem of ’33 base pairs, a terminal ‘hairpin’ loop and two flanking unstructured single-stranded segments. The Pri-miRNA is then processed to a short 70 nucleotide stem-loop structure known as pre-miRNA by a protein complex known as the Microprocessor complex. After the pre-miRNA is generated in the nucleus, it is exported to the cytoplasm by the action of RanGTPase. In the cytoplasm, an RNAse III endonuclease cleaves pre-miRNA into short miRNA duplexes. After cleavage, the miRNA duplec is unwound by an RNA helicase and the mature miRNA strand binds to an Argonaute (Ago) protein into an RNP complex. The mature miRNA binds to the target mRNA and typically in the 3’-untranslated region (3’-UTR), interfering with the translation of the mRNA, thereby, regulating gene expression in cancer by inhibiting translation or by targeting the mRNA for degradation or deadenylation or by upregulating the translation of oncogenes [10,11]. Therefore, MicroRNAs are involved in a variety of biological functions including playing a role in the majority of known hallmarks of cancer, while playing a major role in cancers by functioning as upregulators of oncogenes, or even as tumour suppressors [7]. Furthermore, studies have suggested that due to this, miRNAs have a prognostic value in several human cancers [6,12]. miRNAs can function as both tumour-suppressors as well as oncogenes. Therefore, miRNAs may be capable of offering a sensitive method for HNC detection, monitoring and prognosis. Previous studies have shown that miRNA deregulation occurs in HNC patients. However concrete miRNA prognostic markers capable of informing clinical decision making, have not yet been identified [13]. Therefore, an objective, systematic review on studies about miRNA and HNC was performed, in order to elucidate the significance of specific miRNA in HNC prognosis. Additionally, a comprehensive meta-analysis was performed to investigate the significance of miRNA in determining patient prognosis, with subsequent subgroup analysis further exploring the prognostic significance of frequently expressed miRNAs.

## 2. Methods

### 2.1. Search Strategy and Study Selection

The following systematic review and meta-analysis were conducted based on the Preferred Reporting Items for Systematic reviews and Meta-Analysis (PRISMA) guidelines [14]. The search strategy was designed to be comprehensive and exhaustive with a prime focus on minimising bias and maximising sensitivity. The EMBASE, PubMed, Web of Science and Science Direct bibliographic databases were used to search for relevant published literature in the field of miRNA as prognostic markers in HNC. Predefined ‘search strings’ (as shown in Appendix A) were generated using a few core ‘keywords’, which included;

miRNA

Head and Neck Cancer

Prognosis

Survival

Overall Survival

Disease-Free Survival

miRNA expression

upregulation

downregulation

deregulation/dysregulation

Biomarkers

Treatment

Surgical resection

Radiotherapy

Chemotherapy

Clinical study

Oral cancer

Head and Neck Squamous Cell Carcinoma (HNSCC)

These ‘search strings’ were subsequently used for searching literature databases for relevant studies published until September 2018. The search was conducted individually by two reviewers (C.K. and R.J.), in order to ameliorate any possible selection bias. Initial screening of articles and studies was based on the pertinence of the title and abstract of each publication to the systematic review and meta-analysis. The screening was conducted simultaneously alongside the initial search, at the discretion of the two reviewers (C.K. and R.J.). To further bolster the results of the search strategy, the reference lists of included publications (initial post-screening), were also screened for pertinent studies. Any disagreements arising during initial screening were solved with the inclusion of the third reviewer.

### 2.2. Inclusion and Exclusion Criteria

After initial search and screening, the full-texts of selected studies were then subjected to further rigorous screening based on a set of predefined inclusion and exclusion criteria. This secondary screening process was also conducted by two reviewers in tandem (C.K. and R.J.), based on the following criteria:

Inclusion criteria

The studies discuss miRNA expression in HNC patients.

The studies investigate the association between miRNA expression and patient survival in HNC.

The studies explicitly presented the resulting survival data in terms of hazard ratios (HRs) and 95% confidence intervals (CIs).

The studies assessed patient survival based on the endpoints of OS, disease-free survival (DFS), or both.

The studies provided sufficient data to extract the HR and 95% CI, in case the studies do not explicitly state these values.

Exclusion criteria

Conference abstracts, reviews and correspondence.

Studies that only reported results from in-vitro, in-silico or animal studies.

Studies that were conducted as part of theses, or incomplete studies.

Studies will very small sample sizes, or case studies.

No limits were placed on the patients’ demographic or the clinicopathological characteristics, for the selection of studies for conducting the systematic review and meta-analysis.

### 2.3. Data Extraction

Extraction of data from selected studies was preceded by the preparation of a standardised data extraction form using Microsoft Excel. The data extraction was carried out individually, with all collated data bring combined into one single database at the end of the extraction process, with duplicated data being simultaneously removed. All tables, charts and figures from selected studies were compiled into a separate database for ease of analysis. The following data items were extracted from the studies:

Name of the first author

Year of publication

Country

Number of participants

Study population

Assay methods

Tumour stage

Tumour anatomic location

Clinicopathological characteristics (age, gender, risk factors and metastasis)

Significantly expressed miRNAs

Upregulated, downregulated and dysregulated miRNAs

HR with 95% CI of OS and DFS.

### 2.4. Quality Assessment

The quality of included studies was based on a quality assessment tool developed by the National Heart, Lung and Blood Institute (NHLBI) (USA) for Observational and Cross-sectional studies [15]. This tool was applied to the full texts of all included studies. The NHLBI quality assessment tool uses a set of 14 criteria according to which each study is subjectively rated as ‘good’, ‘fair’, or ‘poor’ based on the opinion of two reviewers. As per the guidelines accompanying the assessment tool, the quality assessment was carried out independently by the reviewers. A ‘high’ quality rating via this tool corresponds to a low risk of bias in the study being assessed, while a ‘low’ quality rating is indicative of a high risk of bias.

### 2.5. Meta-Analysis and Assessment of Heterogeneity

The meta-analysis was conducted using the aid of the Comprehensive Meta-Analysis software [16]. The HR and 95% CI values for OS and DFS extracted from each study were pooled together in the form of a Forest Plot. As HR is the effect size metric chosen for the study, the pooled HR and 95% CI are indicative of the probability of survival for given miRNA expression, thereby indicating the prognostic value of said miRNA. The random-effects model of the meta-analysis was chosen to pool the HR values, due to the inherent heterogeneity that arises from the differences between the study parameters of each study [17]. In the case of studies not presenting the HR and 95% CI values for OS and DFS, the respective values were estimated using the Kaplan-Meier curves for miRNA expression, which were presented in said studies. Additionally, for the assessment and comparison of miRNA concerning each other, each assessment of a discrete patient group within the larger patient group of a single study will be included in the meta-analysis as a unique cohort. Subgroup analysis was carried out based on the survival endpoint of each study (OS or DFS), the change in the miRNA expression (upregulated/downregulated), and the specific miRNA that was represented in at least two separate studies.

The assessment of between-study heterogeneity was based on the Higgins I^2^ statistic, Cochran’s Q and the Tau^2^ value [18,19,20]. The I^2^ statistic was the primary method of assessing heterogeneity, owing to its high power of detection (a lower I^2^ value indicates a lower amount of heterogeneity). However, as I^2^ may generate biased results in a small meta-analysis, the Cochran’s Q Test and Tau^2^ value were also assessed in order to generate a higher degree redundancy in the assessment of heterogeneity between studies [21]. Here, the Tau^2^ value refers to the variance of effect size parameters across the population of studies and reflects the variance of true effect sizes.

### 2.6. Publication Bias

Publication bias is inherent to studies such as systematic reviews and meta-analyses as they consist of previously published studies and literature [22,23,24,25,26,27]. This bias is an extension of the publication process wherein it is more likely that extensive studies and positive results are published, while smaller studies and negative results are unfavoured and often are not published as part of the peer-reviewed literature [23,24,28,29,30,31]. Therefore, publication bias cannot be wholly eradicated from any systematic review and meta-analysis study [32,33]. To ameliorate this issue, assessment of publication bias was conducted to assess the degree of impact publication bias has upon this study’s results [30,34,35,36].

The Egger’s graphical test for assessment of bias was used to construct a funnel plot (a scatter plot constructed using the standard error [*y*-axis] and log (HR) [*x*-axis]), of all included studies. The symmetry of the study distribution on the plot, across the regression line, is inversely correlated with the magnitude of publication bias in the meta-analysis [37].

The Orwin’s Fail-Safe N test was used to determine the presence of missing studies that may skew the regression line in the funnel plot, with Duval and Tweedie’s Trim and Fill method being used for imputation of the missing studies [38,39]. These methods were used in conjunction to adjust the Funnel Plot to better represent the likely publication bias.

Additionally, the Begg and Mazumdar’s Rank Correlation test was used to correlate the ranks of effect sizes and the ranks of their variances, with a positive value indicating a higher test accuracy [40].

## 3. Results

### 3.1. Study Selection and Data Extraction

After following the search strategy, the number of search results of all databases combined totalled to 30,612 potential studies. Out of these studies, a majority were screened out by the reviewers, as they were not relevant to the study being conducted or were focused on tangential topics of research. After this initial screening, 152 studies were selected for further processing. Eliminating duplicates in the search results led to further 22 studies being eliminated, leaving 130 studies for secondary screening based on the inclusion and exclusion criteria. The full texts of the 130 studies were obtained and screened, leading to 99 studies being eliminated and leaving a final total of 31 studies for conducting the study. Screening of the reference lists of these 31 studies revealed 19 more studies that satisfied the requirements for being included in this systematic review and meta-analysis. A final total of 50 studies were included for the systematic review part of the study. However, as not all 50 studies presented sufficient statistical data for conducting of a meta-analysis, studies that did not present HR and 95% CI values, and did not have data from which these values could be extracted, were eliminated from the meta-analysis. Therefore, 16 studies were eliminated, and a total of 34 studies were included in the meta-analysis. The entire process was monitored by the third reviewer at all stages. Figure 1 depicts the entire study selection process in the form of a flowchart.

### 3.2. Study Characteristics

The 50 studies included in this systematic review and meta-analysis study were found to originate from 12 countries around the world, including Brazil (*n* = 3), Canada (*n* = 2), China (*n* = 18), Chile (*n* = 1), Czech Republic (*n* = 1), Denmark (*n* = 1), Germany (*n* = 2), Italy (*n* = 3), Japan (*n* = 5), Taiwan (*n* = 6), UK (*n* = 2) and USA (*n* = 5), while one study did not specify the region in which it was conducted. A total of 6834 patients across the 50 studies were included in this study. Out of all 50 studies, 39 studies were found to involve a higher percentage of men compared to women, with only 1 study reporting having women as the majority of its participants, while the remaining ten studies did not divulge any information regarding the gender ratios of their participants. The miRNA expression in patients was detected using qRT-PCR in 44 studies and immunohistochemistry in 3 studies, with the rest of the studies using other different techniques. In-situ hybridisation (ISH) was also used by three studies, in conjunction with qRT-PCR as the primary method of miRNA quantification. Patients included in 14 of the studies were also found to indulge, or were formerly indulging in smoking, with the studies showing that smokers formed the majority of the at-risk group of these studies, with only three studies describing risk factors besides smoking. The rest of the 33 studies did not provide any information regarding risk factors in the given patient groups. The study and patient characteristics of all 50 studies are described in Table 1.

### 3.3. Meta-Analysis

The primary meta-analysis studied the prognostic significance of 44 miRNA across 34 studies. The meta-analysis was conducted in 2 parts, based on the survival endpoint used by each study. The meta-analysis pooling the HR and 95% CIs of studies that used OS as their endpoint (*n* = 34), gave an overall effect size estimate (HR) of 1.825 (95% CI 1.527–2.181), while the pooled effect size estimate (HR) for studies using DFS as their survival endpoint was 2.596 (95% CI 1.917–3.514). The pooled effect estimate of OS studies (*p* < 0.05) as well as that of DFS studies (*p* < 0.05) were found to be statistically significant, thereby rejecting the null hypothesis. This indicates that any change in miRNA expression (either overexpressed or underexpressed) compared to controls, leads to a lower probability of survival in HNC patients. Additionally the between-study heterogeneity was found to be high, (I^2^ = 75.055; Tau^2^ = 0.228; Cochran’s *Q*= 176.38 for OS meta-analysis and I^2^ = 43.729; Tau^2^ = 0.155; Cochran’s *Q* = 23.103 for DFS). The forest plots of OS and DFS are presented in Figure 2 and Figure 3, respectively.

### 3.4. Subgroup Analysis

#### 3.4.1. Upregulation and Downregulation Subgroups

Subgroups were formed based on whether the miRNAs assessed in each study was found to be upregulated or downregulated compared to controls. The subgroups for upregulated and downregulated miRNA were assessed separately for OS and DFS.

For OS, the upregulated miRNA cohorts (*n* = 25) showed a pooled effect estimate (HR) of 1.762 (95% CI 1.432–2.168; *p* < 0.05), while the downregulated miRNA (*n* = 20) cohorts showed a pooled effect estimate (HR) of 2.018 (95% CI 1.431–2.168; *p* < 0.05). The results were also found to be statistically significant for both upregulation and downregulation.

For DFS, the upregulated miRNA cohort (*n* = 10) showed a pooled effect estimate (HR) of 2.641 (95% CI 1.925–3.623; *p* < 0.05), while the downregulated miRNA cohorts (*n* = 4) showed a pooled effect estimate (HR) of 2.135 (0.730–6.179; *p* > 0.05). Although the pooled effect estimate of studies assessing upregulated miRNA was statistically significant, the same was not observed in the downregulated miRNA studies. For the meta-analysis analysing the effect of downregulated miRNA on DFS, three out of the four cohorts indicated a lower probability of survival and were consistent, while a single study by Ogawa et al [57] was observed to be contradictory to the three studies above, and served to severely skew the overall pooled results. As three of the four studies were consistent with a reduced probability of survival, it is likely that the single study is the outlier, disregarding which, the pooled effect estimate of the three cohorts alone was found to be statistically significant (*p* < 0.05).

#### 3.4.2. miRNA Subgroups

miRNA subgroups were selected based on the miRNA that was frequently represented in multiple cohorts across all studies included in the meta-analysis. miRNA subgroups were also assessed separately based on the survival endpoints (OS and DFS). The miRNA subgroups assessed are miR-21, 155, 200b, 18a, 34c-5p, 125b, 20a, and 375 for the OS group (Appendix A) and miR-21, and 34a for the DFS group (Appendix A).

### 3.5. Overall Survival Group

#### 3.5.1. miR-21

A total of 7 studies assessed miR-21 expression in HNC patients in this group (Figure 4). All seven studies showed that miR-21 is upregulated in HNC patients. The pooled effect size estimate (HR) was found to be 1.591 (1.154–2.194; *p* < 0.05). The pooled effect estimate was statistically significant. Of the seven studies, six showed that upregulated miR-21 expression leads to a lower probability of survival, while a single study presented an outlier result, indicating the opposite. (I^2^ = 48.122; Tau^2^ = 0.085; Cochran’s *Q* = 11.566).

#### 3.5.2. miR-200b

A total of 2 studies assessed miR-200b expression in HNC patients in this group (Figure 5). Both studies showed that miRNA-200b is downregulated in HNC patients. The pooled effect size estimate (HR) was found to be 1.185 (95% CI 0.644–2.182). Both studies individually did not present statistically significant results, and the pooled effect size estimate (HR) was similarly found to be non-significant. (I^2^ = 0.00; Tau^2^ = 0.00; Cochran’s *Q* = 0.292).

#### 3.5.3. miR-155

A total of two studies assessed miR-155 expression in HNC patients (Figure 6). The two studies showed contradicting results regarding miR-155 expression in HNC, with the Hess et al. (2017) study indicating that miR-155 is downregulated in HNC, while the Shi et al. (2015) study claimed that miR-155 is upregulated in HNC. The pooled effect estimate (HR) was found to be statistically significant, with a value of 1.866 (95% CI 1.047–3.326; *p* < 0.05). However, contradicting results of the individual studies limits the applicability of these results to all HNC cases (I^2^ = 0.00; Tau^2^ = 0.00; Cochran’s *Q* = 0.014).

#### 3.5.4. miR-18a

A total of 2 studies assessed miR-18a expression in HNC patients (Figure 7). Both studies indicated that miR-18a is upregulated in HNC. The pooled effect estimate (HR) was found to be statistically significant, with a value of 1.866 (95 % CI 1.047–3.326; *p* < 0.05). (I^2^ = 62.964; Tau^2^ = 0.652; Cochran’s *Q* = 2.7).

#### 3.5.5. miR-34c-5p

A total of 2 studies assessed miR-34c-5p expression in HNC patients (Figure 8). Both studies indicated that miR-34c-5p is downregulated in HNC patients. The pooled effect estimate (HR) was found to be statistically significant, with a value of 4.358 (95% CI 2.376–7.995; *p* < 0.05). (I^2^ = 9.506; Tau^2^ = 0.024; Cochran’s *Q* = 1.105).

#### 3.5.6. miR-125b

A total of 2 studies assessed miR-125b expression in HNC patients (Figure 9). Both studies indicated that miR-125b is upregulated in HNC patients. The pooled effect estimate (HR) was found to be 2.3 (95% CI 0.395–13.397; *p* > 0.05). The results were not statistically significant, although, of the two studies, the Arrigagada et al. (2018) study was found to be statistically significant, indicating miR-125b upregulation leads to a lower probability of patient survival, while the Wilkins et al. (2018) study did not reject the null hypothesis. (I^2^ = 86.696; Tau^2^ = 1.424; Cochran’s *Q* = 7.516).

#### 3.5.7. miR-20a

A total of two studies assessed miR-20a expression in HNC patients (Figure 10). The two studies showed contradicting results regarding miR-20a expression in HNC, with the Chang et al. (2017) study indicating that miR-20a is downregulated in HNC, while the Zeng et al. (2012) study claimed that miR-20a is upregulated in HNC. The pooled effect estimate (HR) was found to be statistically significant, with a value of 4.214 (95% CI 2.165–8.203; *p* < 0.05). However, contradicting results of the individual studies limits the applicability of these results to all HNC cases. (I^2^ = 0.00; Tau^2^ = 0.00; Cochran’s *Q* = 0.524).

#### 3.5.8. miR-375

A total of 3 studies assessed miR-375 expression in HNC patients (Figure 11). Of the three studies, two indicated that miR-375 is upregulated in HNC, while 1 study by Harris et al. (2012) claimed that miR-375 is downregulated in HNC patients. The overall pooled effect estimate (HR) was statistically significant, with a value of 4.482 (95% CI 1.049–19.145). Individually, only the Harris et al. (2012) study results did not reject the null hypothesis, while the remaining two studies individually showed a high degree of statistical significance. Therefore, we may consider the Harris et al. (2012) study as the outlier in this group of studies. (I^2^ = 86.872; Tau^2^ = 1.395; Cochran’s *Q* = 15.235).

### 3.6. Disease-Free Survival Group

#### 3.6.1. miR-21

A total of two studies assessed miR-21 expression in HNC patients in this group (Figure 12). Both studies indicated that miR-21 is upregulated in HNC. The pooled effect size estimate (HR) was found to be 1.466 (95% CI 0.806–2.666; *p* > 0.05). The pooled results were not statistically significant. (I^2^ = 34.5; Tau^2^ = 0.068; Cochran’s *Q* = 1.527).

#### 3.6.2. miR-34a

A total of two studies assessed miR-34a expression in HNC patients in this group (Figure 13). Both studies indicated that miR-34a is downregulated in HNC. The pooled effect size estimate (HR) was found to be 0.190 (95% CI 0.001–130.514; *p* > 0.05). The results were not found to be statistically significant. (I^2^ = 90.317; Tau^2^ = 20.210; Cochran’s *Q* = 10.327).

### 3.7. Publication Bias

The Egger’s graphical test was used to assess for publication bias. The funnel plot was constructed alongside the meta-analysis using the CMA software (Ver 3.3.070, USA). The funnel plots provided in Figure 14 and Figure 15 visually represent the likelihood of publication bias in the OS and DFS groups, respectively, of this systematic review and meta-analysis study. The funnel plot for OS was observed to be slightly asymmetric with a more significant number of studies falling on the right of the line of mean effect. Trim and fill was used to impute for possible missing studies, which led to the imputation of 12 missing studies, adjusting the point estimate and its 95% CI from 1.90043 (1.58306–2.28143) to 1.21935 (1.17451–1.71522), after imputation. Orwin’s Fail-safe N test did not apply to this assessment of publication bias, since the HR of observed studies did not fall between the HRs of the missing studies. Begg and Mazumdar’s rank correlation test presented a Kendall’s Tau value of 0.21919, with continuity correction. For the DFS group, the funnel plot was relatively symmetrical, with the trim and fill required to adjust for two missing studies. Overall, publication bias was not found to have any significant impact on the results of this meta-analysis.

### 3.8. Quality Assessment

The NHLBI Quality Assessment Tool for Observational Cohort and Cross-Sectional Studies was used to assess the quality of included studies. A majority of the studies had good quality scores (38/50), with all the rest having satisfactory scores. However, the core requirement for inclusion into the meta-analysis was based on the availability of good quality statistical data (HR and 95% CI).

## 4. Discussion

This systematic review and meta-analysis were conducted to investigate the prognostic potential of miRNA as biomarkers in HNC, via exploring the association between miRNA expression and survival in HNC patients. The use of miRNAs as prognostic markers in HNC has been the subject of much research, with previous studies reporting that miRNAs have both tumour-suppressing and oncogenic roles and may be either upregulated or downregulated in HNC patients. Therefore, in lieu of miRNAs impact upon cancer progression, they have also been proposed as potential biomarkers predicting patient prognosis. Previous studies have pursued this thread of logic, and have attempted to identify miRNAs that may have the capacity of being utilised as prognostic markers. However, despite this, no new miRNAs have been proposed as prognostic markers in HNC. Furthermore, no previous study has attempted to explore the prognostic potential of multiple miRNAs, as well as explore the impact on HNC based on the magnitude and direction of the deregulation of miRNA expression. This study is the first systematic review and meta-analysis on this topic to have assessed such a wide variety of miRNA and assessed their capacity to act as prognostic markers.

A previous study has also conducted a similar systematic review and meta-analysis [6]. However, the study only focused on miR-21 as a potential prognostic marker in HNC and did not explore the impact or differences of upregulation or downregulation of miRNA on the prognosis of HNC patients. Another previous meta-analysis study was conducted, regarding HNC and miRNA, but focused on the molecular network aspects of miRNA and cancer rather than the clinical utility of miRNA. 

This systematic review and meta-analysis of 50 studies involving 6843 patients, investigating the prognostic value of 43 different miRNA summarises the results via pooled HR values as the effect size estimate of the study. While all miRNA were assessed in a combined meta-analysis to observe their prognostic potential of miRNA as a whole, a few select miRNAs were individually assessed to identify and highlight the specific miRNA which may have potential clinical utility. These miRNA were miR-21, 200b, 155, 18a, 34c-5p, 125b, 20a, 375 and 34a.

While multiple studies were pooled for miR-21 and miR-375, only two cohorts each were pooled for the rest of the miRNAs. miR-21 has been reported as an oncogenic factor and a potential prognostic marker indicating a poor prognosis in multiple types of cancers. Therefore, even though miR-21 is well known for its prognostic potential in cancers, it is not specific to detecting HNC, and is therefore not an ideal miRNA for HNC prognosis in particular. miR-375 is similarly found to be expressed in several cancer types, thereby indicating its lack of specificity. Nevertheless, miR-21 and 375 are dominant miRNA markers for cancer in general and could be considered as prognostic markers for HNC in conjunction with other cancers.

miR-155 and 20a have significant HR values; however, the conflicting reports of the individual studies regarding whether they are upregulated or downregulated hampers their potential for use in the clinical scenario. This uncertainty requires further investigation in longitudinal cohort studies, where the effect of miR-155 and 20a in HNC is further verified, and consistent results are observed.

miR-200b and 125b have effect sizes that are non-significant and do not reject the null hypothesis. This implies that both these miRNAs have a low power of detection, and may not be suitable for use as prognostic markers. However, future studies with larger patient cohorts presenting significant results concerning miR-200b and 125b may serve to highlight the prognostic potential of these miRNA.

miR-18a and 34c-5p are miRNAs offer great potential for prognostic markers in HNC. The effect size values of the pooled results for both miRNAs are significant, as well as high in magnitude, indicating a high power of prognosis. However, as the meta-analysis for these two miRNA was conducted based on the pooling of only two studies each, further verification in future studies is necessary.

This study does have a few limitations [34,91,92,93]. Ideally, meta-analysis should be conducted on all miRNA included in the study. However, this is limited by the quantity of high-quality literature and data that is published in this field. Additionally, this same limitation also makes accounting for other subgroups such as ethnicity, gender and age impossible, as forming these subgroups would fragment the data even further, such that no analysis can be conducted. This meta-analysis was also conducted using HR and 95% CI values extracted from KM curves, which requires approximations to be made during the extraction process, thus introducing some degree of error into the study.

Overall, this systematic review and meta-analysis highlight the miRNA that may have the potential for use as prognostic biomarkers in HNC patients. As the pool of available literature regarding this topic continues to expand, recognizing the influence of miRNAs as a whole and select miRNA specifically may facilitate the transfer of the prognostic value of miRNA from the hypothetical to the clinical sphere of HNC treatment and prognosis.

## 5. Conclusions

The study highlights miR-21, 375, 155, 18a, 34c-5p, 125b, 20a and 375 as miRNA markers that may have the potential for clinical use as prognostic markers in HNC. However, further validation is imperative before we can confirm their utility. Although a few of the miRNA have shown significant results concerning impacting patient survival, the small number of studies that have been pooled to generate the results reduces their applicability in the clinical sphere. Therefore, further large scale and longitudinal patient studies focusing on these miRNA are required.

## Figures and Tables

**Figure 1 cells-08-00772-f001:**
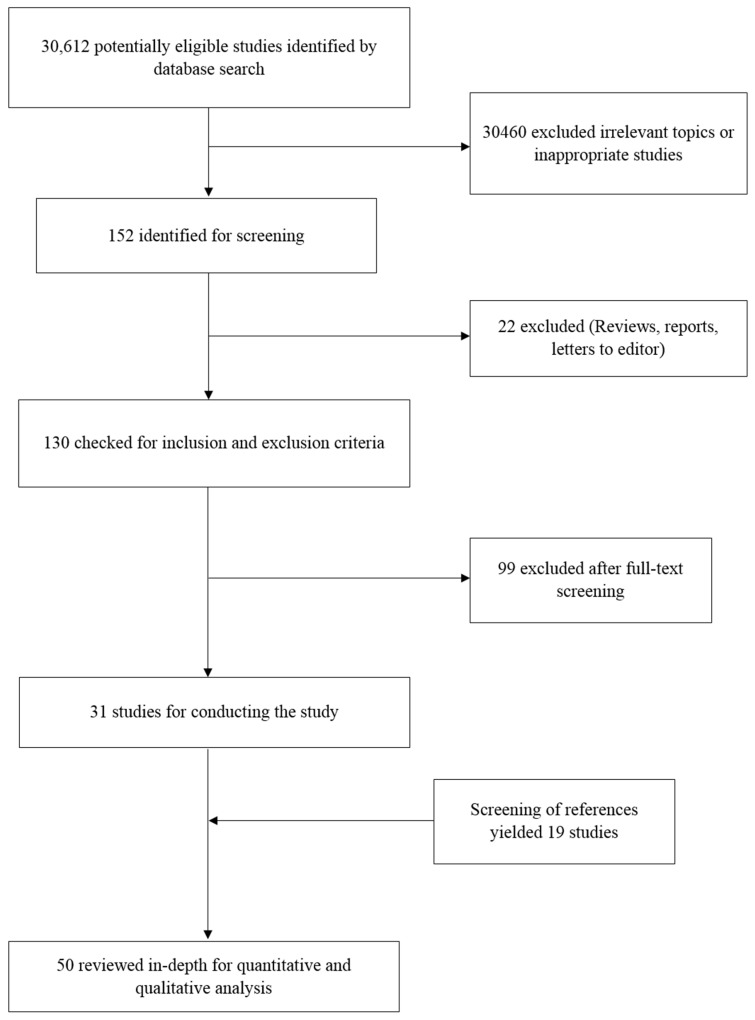
Flow chart describing search strategy.

**Figure 2 cells-08-00772-f002:**
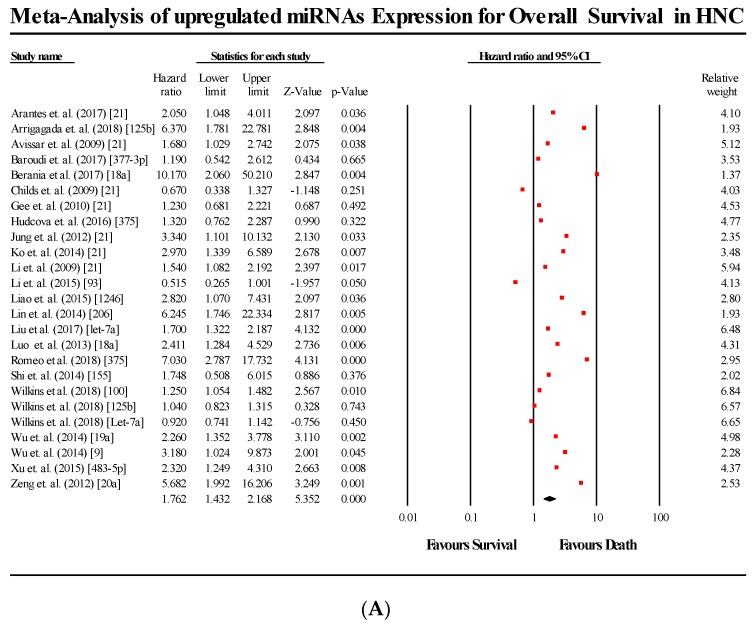
Forest plots for the miRNAs associated with OS. (**A**) Upregulated miRNA; (**B**) Downregulated miRNA.

**Figure 3 cells-08-00772-f003:**
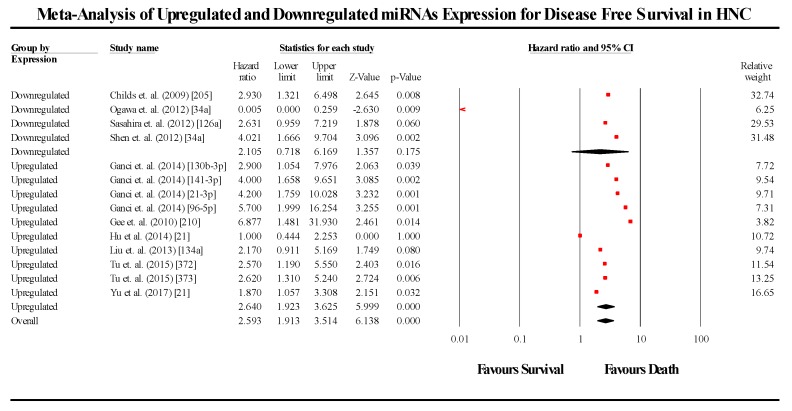
Forest plot for the miRNAs associated with DFS.

**Figure 4 cells-08-00772-f004:**
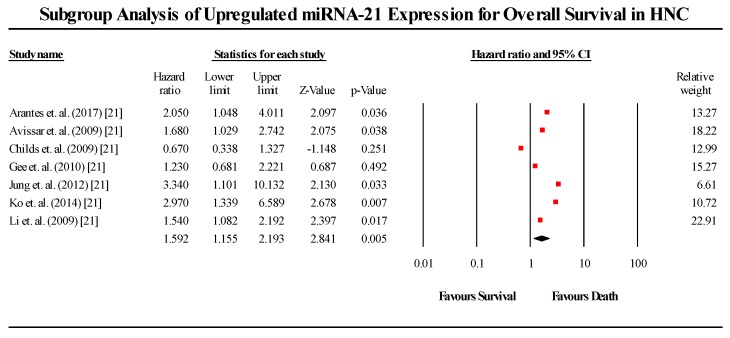
Forest plot for miR-21 association with OS.

**Figure 5 cells-08-00772-f005:**
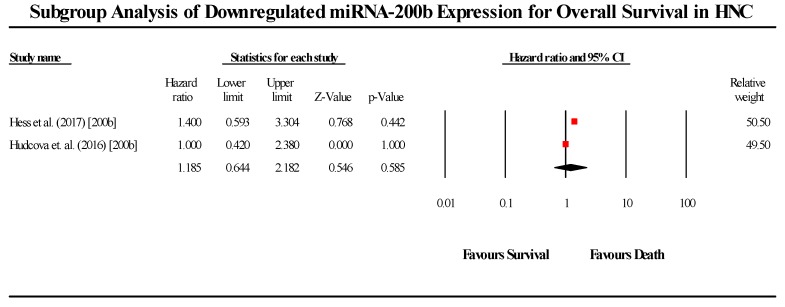
Forest plot for miR-200b association with OS.

**Figure 6 cells-08-00772-f006:**
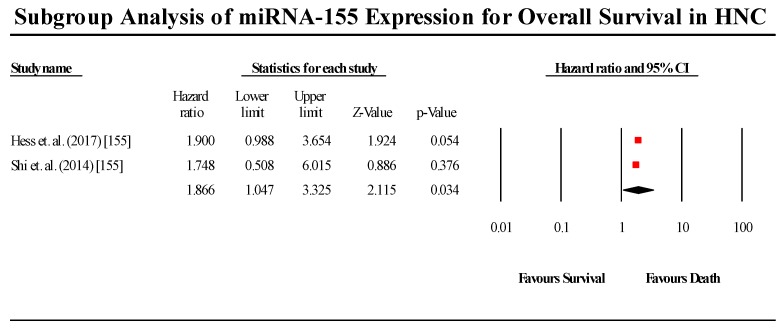
Forest plot for miR-155 association with OS.

**Figure 7 cells-08-00772-f007:**
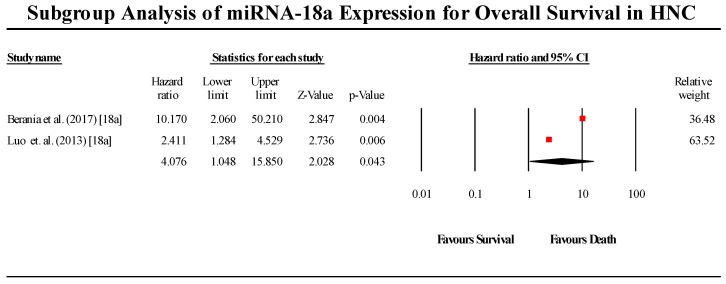
Forest plot for miR-18a association with OS.

**Figure 8 cells-08-00772-f008:**
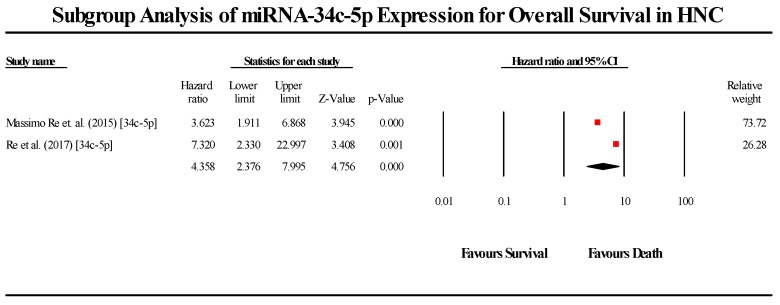
Forest plot for miR-34c-5p association with OS.

**Figure 9 cells-08-00772-f009:**
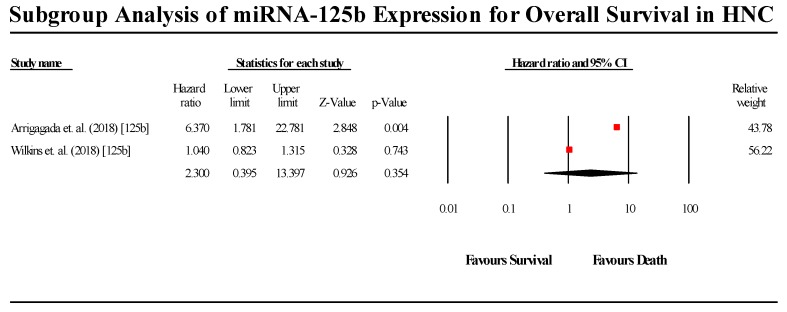
Forest plot for miR-125b for association with OS.

**Figure 10 cells-08-00772-f010:**
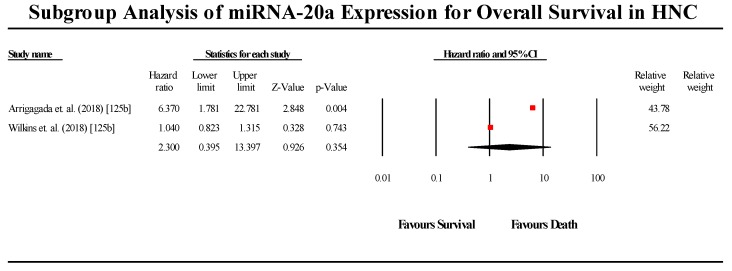
Forest plot for miR-20a association with OS.

**Figure 11 cells-08-00772-f011:**
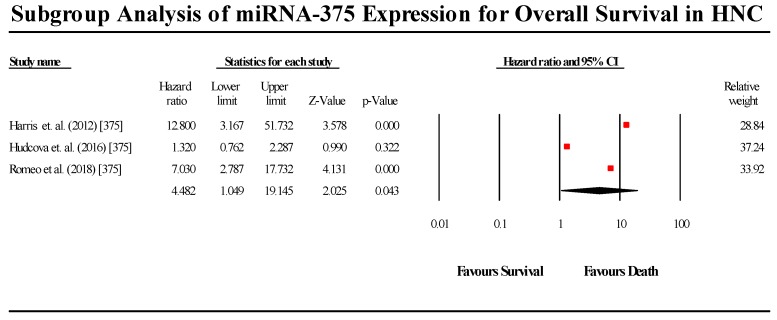
Forest plot for miR-375 association with OS.

**Figure 12 cells-08-00772-f012:**
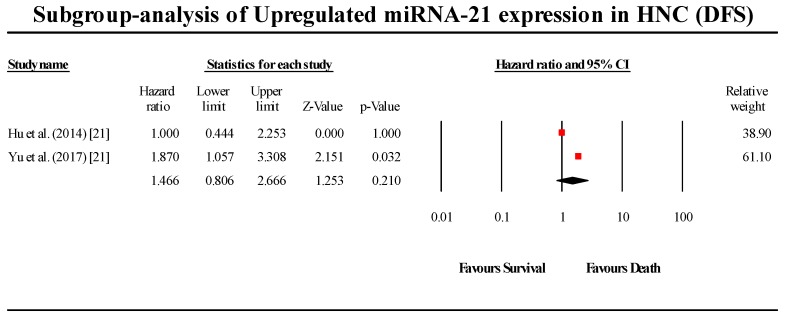
Forest plot for miR-21 association with DFS.

**Figure 13 cells-08-00772-f013:**
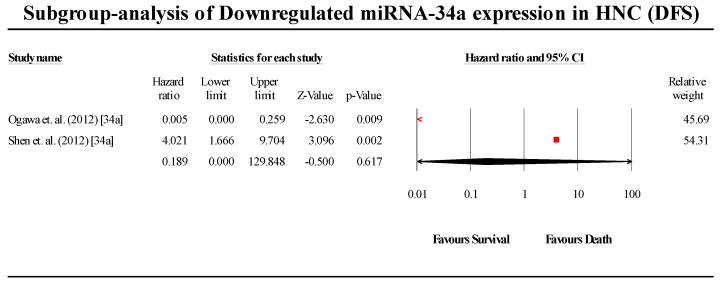
Forest plot for miR-34a association with DFS.

**Figure 14 cells-08-00772-f014:**
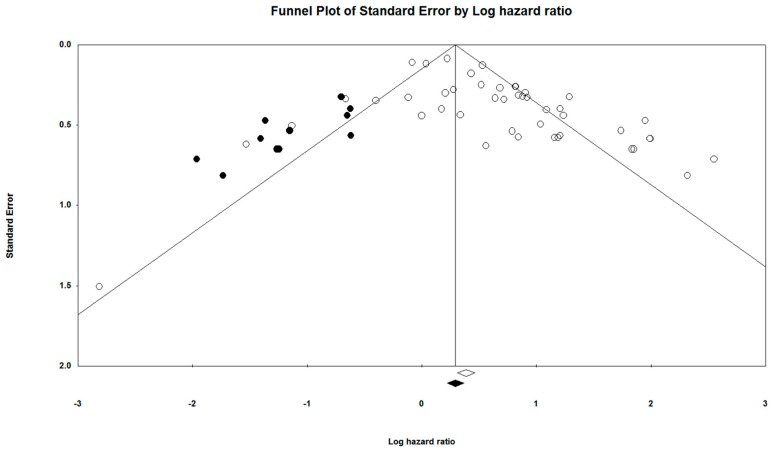
Funnel plot for the studies prognostic of OS.

**Figure 15 cells-08-00772-f015:**
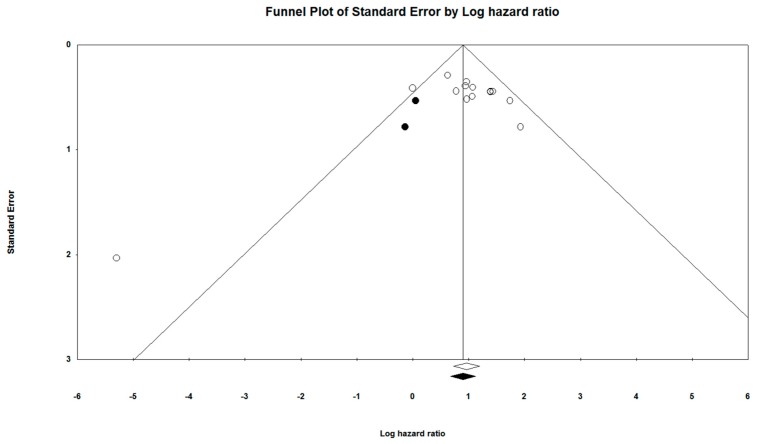
Funnel plot for the studies prognostic of DFS.

**Table 1 cells-08-00772-t001:** The study and patient characteristics of all 50 studies.

Article	Year	miR	Sample Size	Anatomic Location	Assay Method	Study Population	Gender	Stage	Metastasis	Risk Factors	Age
Carvalho et al. [41]	2015	miR-203 miR-205	127	Tongue 58.3% Floor of mouth 31.3% Alveolar ridge 8.3% Lower gum 2.1%	qRT-PCR	Brazil	Male (79.2%)	T2 (62.5%)	Metastases + ve (52.083%) Metastases − ve (47.916%)	Smoking (47.9%)	43–84
Hou et.al. [42]	2015	miR-223 miR-99a miR-21	16	Head and Neck	qRT-PCR	Japan	Male (93.75%)	T2 (18.75%) T3 (12.5%) T4 (68.75%)	NA	NA	48–80
Maia et al. [43]	2015	miR-296-5p	34	Supraglottic 20.6% Glottic 79.4%	qRT-PCR	Brazil	Male (88.2%)	T1 (47.1%) T2 (52.9%)	NA	Tobacco (91.2%)	≤60 years (47%) >60 years (53%)
Hudcova et al. [44]	2016	miR-29c miR-200b miR-375	42	Head and Neck	qRT-PCR	Czech Republic	Male (100%)	T1 + T2 (45%) T3 + T4 (55%)	Metastases + ve (11.42%) Metastases − ve (88.57%)	NA	NA
Wang et al. [45]	2015	miR-451	50	Head and Neck	qRT-PCR	China	NA	NA	NA	NA	NA
Arantes et al. [46]	2017	miR-21	71	Oropharynx 49.3% Larynx 39.4% Hypopharynx 11.3%	qRT-PCR	Brazil	Male (95.8%)	T2 + T3 (64.8%) T4 (35.2%)	NA	HPV (8.45%) Tobacco (80.3%) Alcohol (38.0%)	40–76
Xu et al. [47]	2015	miR-483-5p	101	Oral Cavity	qRT-PCR	China	Male (76.2%)	T1 + T2 (50.5%) T3 + T4 (49.5%)	NA	Smoking (72.3%) Alcohol (68.3%)	53.2 ± 10.3
Li et al. [48]	2015	miR-93	103	Supraglottic 25.24% Glottic 55.33% Hypopharynx 9.7% Oral Cavity 9.7%	ISH, qRT-PCR	China	Male (96.1%)	T1 (15.5%) T2 (35%) T3 (40.8%) T4 (8.7%)	Metastases + ve (38.83%)	NA	<58 (46%) ≥58 (54%)
Hu et al. [49]	2014	miR-21 miR-375	46	Glottic 71.7% Supraglottic 23.9% Subglottic 4.4%	qRT-PCR	China	Male (91.3%)	T0 + T1 + T2 (45.7%) T3 + T4 (54.3%)	NA	Smoking (72.1%) Alcohol (46.3%)	59.2 ± 7.84
Hedback et al. [50]	2014	miR-21	86	Oral Cavity	ISH, Immunohistochemistry	Denmark	NA	NA	NA	NA	NA
Sun et al. [51]	2015	miR-320a	450	Salivary Gland	ISH, Immunohistochemistry	China	Male (47.56%)	T1 + T2 (65.33%) T3 + T4 (34.67%)	Metastases + ve (43.56%)	NA	<50 (49%) ≥50 (51%)
Saito et al. [52]	2013	miR-196a	84	Larynx	qRT-PCR	Japan	NA	NA	NA	NA	NA
Li et al. [53]	2009	miR-21	103	Tongue	qRT-PCR	China	Male (54.36%)	T1 + T2 (58.25%) T3 + T4 (41.75%)	Metastases + ve (27.18%)	NA	<50 (46%) ≥50 (54%)
Liu et al. [54]	2012	miR-93 miR-142-3p miR-29c miR-26a miR-30e	465	Nasopharyngeal	qRT-PCR	China	Male (74.19%)	T1 (21.94%) T2 (27.31%) T3 (23.66%) T4 (27.10%)	Metastases + ve (19.78%))	NA	47.09 ± 11
Summerer et al. [55]	2013	miR-425-5p miR-21-5p miR-106b-5p miR-93-5p	18	Larynx 27.77% Oropharynx 16.66% Mouth floor 11.11% Tongue 11.11% Esophagus 5.55% Hypopharynx 5.55% Maxilla 5.55% Nasopharyngeal 5.55% Sinuses 5.55% Soft palate 5.55%	qRT-PCR	Germany	Male (77.78%)	T1 (22.22%) T2 (11.11%) T3 (33.33%) T4 (33.33%)	Metastases + ve (11.11%)	NA	45.1–80.6
Suh et al. [56]	2015	miR-196a	16	Oral Cavity	qRT-PCR	UK	NA	NA	NA	NA	NA
Ogawa et al. [57]	2012	miR-34a	24	Sinonasal	miRNA-Microarray	Japan	Male (66.67%)	T2 (4%) T3 (41.67%) T4 (54.17%)	Metastases + ve (8.33%)	NA	>60 (59%) <60 (41%)
Avissar et al. [58]	2009	miR-375 miR-21	169	Oral 64% Pharynx 21% Larynx 15%	qRT-PCR	USA	Male (68%)	T1 + T2 (28%) T3 + T4 (72%)	NA	HPV (17.4%) Alcohol (88.5%) Smoking (84.5%)	61.5 ± 11.9
Massimo Re et al. [59]	2015	miR-34c-5p	90	Supraglottic 21.1% Transglottic 73.3% Subglottic 5.6%	qRT-PCR	Italy	Male (96.6%)	T3 (66.7%) T4 (33.3%)	Metastases + ve (0%)	NA	66.51 ± 8.02
Sun et al. [60]	2013	miR-363	62	Tongue 41.9% Gingival 21% Cheek 11.3% Floor of Mouth 17.7% Oropharynx 8.1%	qRT-PCR	China	Male (69.4%)	T1 + T2 (43.5%) T3 + T4 (36.5%)	Metastases + ve (54.83%)	Smoking (48.4%) Drinking (32.3%)	≥60 (42%) <60 (58%)
Tian et al. [61]	2014	miR-203	56	Glottic 53.57% Supraglottic 46.43%	qRT-PCR	China	Male (71.43%)	T1 + T2 (42.85%) T3 + T4 (57.14%)	Metastases + ve (50%)	NA	≥59 (57%) <59 (43%)
Chang et al. [62]	2013	miR-17 miR-20a	98	Buccal Mucosa 43.88% Tongue 29.59% Gingiva 21.43% Floor of Mouth 5.10%	qRT-PCR	Taiwan	Male (84.7%)	T1 + T2 (44.9%) T3 + T4 (55.1%)	Metastases + ve (37.75%)	Smoking (82.65%)	>50 (35%) <50 (65%)
Gee et al. [63]	2010	miR-210	46	Oral Cavity 21% Oropharynx 46% Hypopharynx 19% Larynx 11% Paranasal Sinus 2%	qRT-PCR	UK	Male (80.43%)	T1 (10.87%) T2 (30.43%) T3 (15.22%) T4 (43.48%)	NA	Smoking (86.96%) Alcohol (78.26%)	43–92
Lenarduzzi et al. [64]	2013	miR-193b	51	Head and Neck	qRT-PCR	Canada	NA	NA	NA	NA	NA
Childs et al. [65]	2009	miR-205 Let-7d miR-21	104	Oral Cavity 30% Oropharynx 46% Hypopharynx 9% Larynx 31%	qRT-PCR	US	Male (68%)	T1 + T2 (23%) T3 + T4 (77%)	NA	Smoking (82%) HPV (36%)	<60 (40%) >60 (61%)
Shen et al. [66]	2012	miR-34a	69	Larynx	qRT-PCR	China	NA	T1 + T2 (60.87%) T3 + T4 (39.13%)	Metastases + ve (34.78%)	NA	<60 (48%) ≥60 (52%)
Luo et al. [67]	2013	miR-18a	168	Nasopharyngeal	qRT-PCR	China	Male (75.6%)	T1 + T2 (42.86%) T3 + T4 (57.14%)	Metastases + ve (64.88%)	NA	≥50 (59%) <50 (41.%)
Jung et al. [68]	2012	miR-21	17	Tongue 94.12% Oropharynx 5.88%	qRT-PCR	USA	NA	NA	NA	HPV (58.82%)	41–69
Sasahira et al. [69]	2012	miR-126a	118	Tongue 54.24% Other 45.76%	qRT-PCR	Japan	Male (57.63%)	T1 + T2 (76.27%) T3 + T4 (23.73%)	Metastases + ve (28.81%)	NA	≤65 (39%) >65 (61%)
Liu et al. [70]	2014	miR-134a	96	Buccal Mucosa 35.41% Tongue 27.08% Oral pharynx 37.5%	qRT-PCR	Taiwan	Male (93.75%)	T1 + T2 + T3 (28.12%) T4 (71.88%)	Metastases + ve (6.25%)	NA	53.5 (Average)
Shi et al. [71]	2014	miR-155	30	Oral Cavity	qRT-PCR, FISH	China	Male (63.33%)	T1 (10%) T2 (16.67%) T3 (33.33%) T4 (40%)	NA	Smoking (46.67%) Alcohol (53.33%)	56.4 ± 8.6 (40-75)
Harris et al. [72]	2012	miR-375	123	Oral Cavity 35% Oropharynx 30% Larynx 35%	qRT-PCR	US	Male (69.1%)	T1 + T2 (19.5%) T3 + T4 (80.5%)	NA	Smoking (60.9%) Alcohol (27.6%) HPV (25.2%)	≤58 (37%) 59–66 (31%) ≥67 (33%)
Huang et al. [73]	2014	miR-491-p5	33	Oral Cavity	qRT-PCR, FISH	Taiwan	Male (96.9%)	T1 (9.1%) T2 (51.5%) T3 (3.0%) T4 (36.4%)	NA	NA	≤60 (21%) >60 (79%)
Shiiba et al. [74]	2013	miR-125b	50	Oral Cavity	qRT-PCR	Japan	NA	T1 (10%) T2 (12%) T3 (14%) T4 (64%)	NA	NA	NA
Zeng et al. [75]	2012	miR-20a	160	Nasopharyngeal	qRT-PCR	China	Male (61.25%)	T1 (1,25%) T2 (15.63%) T3 (34.38%) T4 (40%)	NA	NA	46.41 ± 10.74
Liu et al. [76]	2013	miR-451	280	Nasopharyngeal	qRT-PCR	Taiwan	Male (73.57%)	T1 + T2 (50.71%) T3 + T4 (49.28%)	NA	NA	≤45 (49%) >45 (51%)
Yang et al. [77]	2011	miR-181a	39	Oral Cavity	qRT-PCR	Taiwan	Male (44.87%)	T1 + T2 + T3 (33.33%) T4 (66.66%)	NA	NA	NA
Wu et al. [78]	2014	miR-19a	83	Laryngeal	qRT-PCR	China	Male (68.67%)	NA	Metastases + ve (34.93%)	NA	≥56 (51%) <56 (49%)
Peng et al. [79]	2014	Let-7g miR-125b miR-218	29	Oral Cavity	qRT-PCR	Taiwan	NA	NA	NA	NA	NA
Arriagada et al. [80]	2018	miR-215b	32	Head and Neck	qRT-PCR	Chile	Male (55.9%)	T1 + T2 (75.2%) T3 + T4 (47.7%)	NA	Smoking (62.5%) Drinking (50.5%)	<64 (86%) ≥64 years (44%)
Baroudi et al. [81]	2017	miR-377-3p	199	Larynx 31% Oral cavity 64% Oropharynx 5%	GSEA	NA	Male (28%)	T1 (9%) T2 (18%) T3 (27%) T4 (53%)	NA	Smoking (52%) Alcohol (66%)	≤ 70 years (80%) > 70 years (20%)
Berania et al. [82]	2017	miR-18a miR-548b	58	Oral tongue squamous cell carcinoma	qRT-PCR	Canada	Male (71%)	NA	NA	Smoking (72%) Drinking (41%) HPV (22%)	≤ 50 (28%) > 50 (72%)
He et al. [83]	2017	miR-300	133	Laryngeal squamous cell carcinoma	qRT-PCR	China	Male (65%)	T1 + T2 (50%) T3 + T4 (50%)	Metastasis + ve (55%)	NA	<50 (35%) ≥50 (65%)
Hess et al. [84]	2017	miR-200b miR-155 miR-146a	149	Oropharynx 52% Hypopharynx 48%	qRT-PCR	Germany	NA	NA	NA	NA	NA
Jiang et al. [85]	2017	miR-212	73	Nasopharyngeal	qRT-PCR	China	Male (59%)	T1 + T2 (34%) T3 + T4 (66%)	Metastasis + ve (56%)	NA	≤45 (48%) >45 (52%)
Liu et al. [86]	2017	let-7a	131	Thyroid	qRT-PCR	China	Male (33%)	T1 + T2 (39%) T3 + T4 (61%)	Metastasis + ve (53%)	NA	< 45 (44%) ≥ 45 (56%)
Re et al. [87]	2017	miR-34c-5p	43	Supraglottic (18.60%) Transglottic (76.74%) Subglottic (4.65%)	qRT-PCR	Italy	Male (97.67%)	T3 (72%) T4 (28%)	Metastasis + ve (0%)	NA	66.51 ± 8.02
Romeo et al. [88]	2018	miR-375	36	Medullary thyroid	qRT-PCR	Italy	Male (58.3%)	T1 + T2 (25%) T3 + T4 (63.8%)	Metastasis + ve (72.2%)	NA	Mean 55.5
Wilkins et al. [89]	2018	miR-100 miR-125b Let-7a	2083	Oral cavity (31.7%) Pharynx (52.7%) Larynx (15.6%)	Axiom miRNA Target Site Genotyping Array	USA	Male (24.5%)	T1 + T2 (25.9%) T3 + T4 (74.1%)	NA	Smoking [current] (25.9%) Smoking [former] (42.9%)	≤50 (24.8%) >50 to ≤60 (36.2%) >60 to ≤70 (25.9%) >70 (13.1%)
Yu et al. [90]	2017	miR-21	100	Buccal mucosa (37%) Tongue (35%) Mouth floor (12%) Others (16%)	Immunohistochemistry	China	Male (92%)	T1 + T2 (23%) T3 + T4 (77%)	Metastasis + ve (28%)	NA	≤55 (56%) >55 (44%)

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
