# Peer review of "Prognostic Value of miRNAs in Head and Neck Cancers: A Comprehensive Systematic and Meta-Analysis"

_cells, 2019, doi:10.3390/cells8080772_

Round 1
Reviewer 1 Report
Dear authors
I have reviewed the revised version of the manuscript, and I am happy with the additional observations and comments included, about the mechanism of action of miRNAs. However, I am not convinced about the lack of importance of the expression levels of miRNAs for your study, together with the claim about the non-invasiveness of the miRNAs as biomarkers. I agree that there are a few similar studies in the literature, and it is good that the authors considered that.
Best regards
Reviewer 2 Report
Kumarasamy et al. addressed all my concerns. I have no further questions or concerns to add and I feel the manuscript is both suitable with the scope of the journal and the results are solid.
This manuscript is a resubmission of an earlier submission. The following is a list of the peer review reports and author responses from that submission.
Round 1
Reviewer 1 Report
In this manuscript Kumarasamy et al. the prognostic value of miRNA in head and neck cancers. Some concerns and remaining questions have to be addressed to improve the quality of the manuscript.
1. It is now established that the secondary structure of the 5′ untranslated region (5′ UTR) of messenger RNA is important for microRNA-mediated gene regulation in humans. Please add this important information in the introduction.
2. There are serious problems with bibliographic references. For example, refs 56 and 84 are from the same group, the format in the text is different. authors should check carefully the bibliography.
3. In addition, I wonder if the same cohort was used in these studies of the group of Olivieri (réf 56 et 84), please clarify this point.
4. The authors should improve the quality of the figures and homogenize them.
5. The authors should not report not significant results as for miR-34a.
6. The authors should separate Figure 2 into 2 figures (i.e. down and up-regulated mi-RNA).
Reviewer 2 Report
The paper by Kumarasamy and coworkers, proposed a meta-analysis to determine the possible use of miRNAs as prognostic biomarkers in head and neck carcinomas. The paper is exhaustive in the number of datasets analyzed and the experimental analysis was well performed.
However, there are some important technical concerns in the design of the study that in my opinion prevent the acceptance of the article in this current form.
The main problem of the study design is related with the concept of biomarker itself. The study combines papers and data that are not comparable. In fact at least three of the referred studies were performed by collecting and measuring the miRNA levels in biofluids, namely in plasma or serum : Xu et al, Tumor Biology 2016, 37, 447-453; Summerer et al, Radiation oncology 2013, 8, 296; and Zeng et al, PLoS One 2012, 7, e46367. The authors cannot compare the miRNA levels in solid tissues vs biofluids, even in a meta-analysis study, since the origin of circulating miRNAs is not always an isolated tissue, involving many organs and cell types. It is not always true that the source of circulating miRNAs is a tumor, rather a combination of tissues and cells, that could respond to long-range metabolic interactions. Moreover, the authors claimed that the use of miRNAs could be a good protocol for “non-invasive” diagnosis, but most of the referred papers in the meta-analysis are based on the collection of tumor biopsies.
Minor comments:
1.- The authors should complete the definition and biological implication of miRNAs. For instance, the text stated that “MicroRNAs are small non-coding RNAs (~22 nucleotides) transcribed from DNA into RNA hairpins”. In fact, this is a very limited and simplistic approach to the definition of what we can consider to be a miRNA. MiRNAs are small non-coding RNAs, that are generated by the transcription of specific genomic units that will generate first a long primary miRNA, that contains the characteristic hairpin-like structure, further processed into a precursor miRNA and the mature form of the functional miRNA. Please, complete the text also with some details of the mechanism of action of miRNAs and what could be the implications of their action in the context of cancer (maybe citing 2-3 references, that could be reviews).